# Luna: Linear Unified Nested Attention

**Xuezhe Ma**[*]
ISI, USC
xuezhema@isi.edu

**Xiang Kong**[*]
LTI, CMU
xiangk@cs.cmu.edu

**Sinong Wang**[*]
Facebook AI
sinongwang@fb.com

**Chunting Zhou**
LTI, CMU
chuntinz@cs.cmu.edu

**Jonathan May**
ISI, USC
jonmay@isi.edu

**Hao Ma, Luke Zettlemoyer**
Facebook AI
{haom, lsz}@fb.com

## Abstract

The quadratic computational and memory complexities of the Transformer's attention mechanism have limited its scalability for modeling long sequences. In this paper, we propose Luna, a linear unified nested attention mechanism that approximates softmax attention with *two nested linear attention functions*, yielding only linear (as opposed to quadratic) time and space complexity. As compared to a more traditional attention mechanism, Luna introduces an additional sequence with a fixed length as input and an additional corresponding output, which allows Luna to perform attention operation linearly, while also storing adequate contextual information. We perform extensive evaluations on three benchmarks of sequence modeling tasks: long-context sequence modeling, neural machine translation and masked language modeling for large-scale pretraining. Competitive or even better experimental results demonstrate both the effectiveness and efficiency of Luna compared to a variety of strong baseline methods including the full-rank attention and other efficient sparse and dense attention methods. The implementation of our model is available at https://github.com/XuezheMax/fairseq-apollo.

## 1 Introduction

Transformers (Vaswani et al., 2017) are surprisingly versatile models that preform well on a wide range of language and vision tasks, including machine translation (Vaswani et al., 2017; Ott et al., 2018), language understanding (Devlin et al., 2019), image recognition (Dosovitskiy et al., 2020) and bioinformatics (Madani et al., 2020). Attention (Bahdanau et al., 2015) provides the key mechanism that captures contextual information from the entire sequence by modeling pairwise interactions between the inputs at every timestep. However, a common weakness of Transformers is their quadratic time and memory complexity within the attention mechanism w.r.t the length of the input sequence, which prohibitively restricts their potential application to tasks requiring longer input sequences.

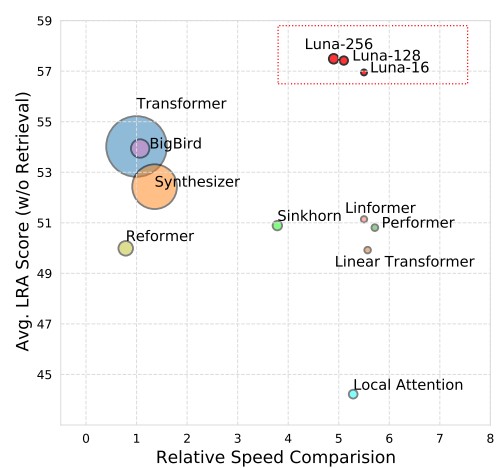

Figure 1: Trade-off between accuracy (y-axis), speed (x-axis) and memory (cir-radius) on LRA.

A number of techniques have been recently introduced to improve the time and memory efficiency of Transformer models (*'xformers'*) (Tay et al., 2020b, 2021). One popular technique is using sparsity to restrict the attention field range, such as local attention (Parmar et al., 2018), blockwise attention (Qiu et al., 2019), strided attention patterns (Child et al., 2019; Beltagy et al., 2020),

---

[*]Equal contribution.

35th Conference on Neural Information Processing Systems (NeurIPS 2021).

compressed attention (Liu et al., 2018), and attention with learnable patterns (Kitaev et al., 2020; Tay et al., 2020a; Roy et al., 2021). Another emerging approach is to improve efficiency by leveraging low-rank approximations of the attention matrix. Linformer (Wang et al., 2020), for example, projects the length dimension of key and value matrices to a fixed-dimensional representation by assuming low-rank structure in the full-rank attention matrix. Recently, some kernel-based methods, such as Linear Transformer (Katharopoulos et al., 2020), Performer (Choromanski et al., 2020) and Random Feature Attention (Peng et al., 2021), attempt to efficiently approximate regular (softmax) full-rank attention through kernelization. Although these models demonstrate better *asymptotic* complexity for long sequences, their efficiency gains are less prominent for moderate length sequences and their performance remains behind Transformers with regular attention.

In this work, we propose *a linear unified nested attention mechanism* (**Luna**), which uses two nested attention functions to approximate the regular softmax attention in Transformer (§2). Specifically, with the first attention function, Luna packs the input sequence into a sequence of fixed length. Then, the packed sequence is unpacked using the second attention function (§3.1). As compared to a more traditional attention mechanism, Luna introduces an additional sequence with a fixed length as input and an additional corresponding output. Importantly, the extra input allows Luna to perform attention operation linearly as efficiently as Linformer (Wang et al., 2020), while also storing adequate contextual information. Unlike Linformer, Luna is capable of modeling variable-length sequences and autoregressive (causal) attention (§3.4). We perform extensive experiments on three sequence modeling tasks, including long-context sequence modeling, neural machine translation, and masked language modeling for large-scale pretraining and downstream task finetuning. Compared to a variety of strong baseline models, Luna achieves competitive or even better performance, while acquiring prominent gains of efficiency in both speed and memory (see Figure 1). More importantly, Luna manages to obtain superior performance with small projection lengths such as 16 (§4).

## 2  Background

### 2.1  Attention

The traditional attention mechanism is a function:

$$Y = \text{Attn}(X, C) = \omega\left(\frac{XW_Q(CW_K)^T}{\sqrt{d}}\right)CW_V \tag{1}$$

where the attention function $\text{Attn} : \mathbb{R}^{n \times d} \times \mathbb{R}^{m \times d} \to \mathbb{R}^{n \times d}$ takes as inputs two sequences: the query sequence $X \in \mathbb{R}^{n \times d}$ with length $n$ and the context sequence $C \in \mathbb{R}^{m \times d}$ with length $m$, and output one sequence $Y \in \mathbb{R}^{n \times d}$ with the same length $n$ as the query $X$. $d$ is the embedding dimension, and $W_Q$, $W_K$, $W_V \in \mathbb{R}^{d \times d}$ are three learnable parameters that project the input sequences into the space of query, key and value matrices: $Q = XW_Q$, $K = CW_K$, $V = CW_V$. $\omega$ is an activation function, e.g. the *softmax* function in regular attention. Note that the formulation in (1) is applicable to both *cross-attention* where $C$ and $X$ are the representations from Transformer encoder and decoder, respectively, and *self-attention* where $X$ and $C$ are the same sequence ($X = C$). In practice, the multi-head variant of attention (Vaswani et al., 2017), which performs the attention function $h$ times in parallel, is commonly used. Throughout this paper, we omit $h$ for simplicity.

In particular, the matrix $A = \omega\left(\frac{QK^T}{\sqrt{d_k}}\right) \in \mathbb{R}^{n \times m}$ in (1) is called the attention matrix which specifies the alignment scores between every pair of tokens in sequences of queries $X$ and contexts $C$. Calculating $A$ takes $O(nm)$ time and space, which is quadratic with respect to the sequence length and becomes a significant bottleneck when processing long sequences.

### 2.2  Transformer Layers

The other two key components of Transformer, besides attention, are position-wise feed-forward networks (FFN) and layer normalization (Ba et al., 2016). Technically, the position-wise feed-forward layer operates on each position independently and layer normalization plays a crucial role in controlling the gradient scales (Xiong et al., 2020). Each Transformer layer can be expressed as:

$$\begin{aligned} X_A &= \text{LayerNorm}(\text{Attn}(X, C) + X) \\ X' &= \text{LayerNorm}(\text{FFN}(X_A) + X_A) \end{aligned} \tag{2}$$

where $X$ and $C$ are the two input sequences and $X'$ is the output of the Transformer layer. The Transformer layer in (2) adopts the original post-layer normalization architecture (Vaswani et al., 2017; Devlin et al., 2019) that places layer normalization after residual connection, rather than pre-layer normalization (Vaswani et al., 2018; Wang et al., 2019).

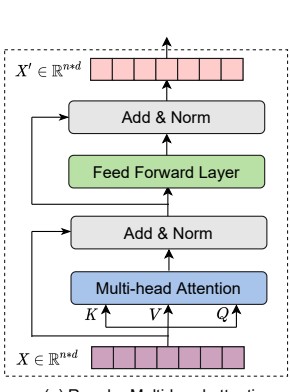 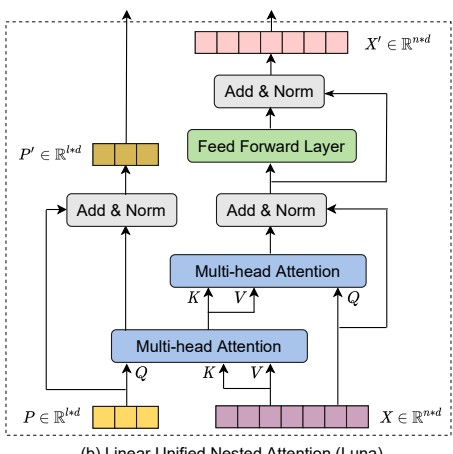

(a) Regular Multi-head attention

(b) Linear Unified Nested Attention (Luna)

Figure 2: Illustration of the architecture of one Transformer encoder layer (left) versus one Luna encoder layer (right).

# 3 Linear Unified Nested Attention (Luna)

Our goal is to design an efficient attention mechanism to solve the quadratic complexity problem of full attention. We first introduce the proposed *linear unified nested attention* mechanism, named *Luna attention* (§3.1), and the architecture of each Luna layer (§3.2). Then, we present the variant of Luna for causal attention, named *Luna causal attention* (§3.3). Finally, we discuss the differences between Luna and three closely related models: Linformer (Wang et al., 2019), Set Transformer (Lee et al., 2019) (§3.4) and Shared Workspace (Goyal et al., 2021).

## 3.1 Pack and Unpack Attention

The key idea behind Luna is to decouple the regular attention function in (1) into two nested attention operations, both of which have linear efficiency. To achieve this, besides the original query and context input sequences, Luna introduces an extra input that is a sequence with fixed (constant) length. With this extra input as the query sequence, Luna uses its first attention, named *pack attention*, to pack the context sequence into a fixed-length sequence. Formally, let $P \in \mathbb{R}^{l \times d}$ denote the extra input sequence with fixed length $l$. The pack attention first packs $C$ to $Y_P$ with $P$ as the query sequence:

$$Y_P = \text{Attn}(P, C) \tag{3}$$

where $\text{Attn}(\cdot, \cdot)$ is the regular attention function in (1), $C \in \mathbb{R}^{m \times d}$ is the context sequence, and $Y_P \in \mathbb{R}^{l \times d}$ is the output of the pack attention, which is named the *packed context*. Since the length of $P$ is a constant $l$, the complexity of pack attention is $O(lm)$, which is linear with respect to $m$.

To unpack the sequence back to the length of the original query sequence $X$, Luna leverages its second attention, named *unpack attention*:

$$Y_X = \text{Attn}(X, Y_P) \tag{4}$$

where $X \in \mathbb{R}^{n \times d}$ is the original query sequence. Similar to pack attention, the complexity of unpack attention is $O(ln)$, which is also linear with repect to $n$.

**Encoding Contextual Information in $P$.** The next question is where the extra input sequence $P$ comes from. One straightforward choice is to format $P$ as a learnable parameter of each Luna layer. One obvious drawback of this method, however, is that $P$ would not capture any contextual information. To enhance the capacity of the Luna model, we propose to formulate $Y_P$ as an additional output of each Luna layer, corresponding to $P$. Formally, the Luna attention function $\text{LunaAttn}(\cdot, \cdot, \cdot)$ takes three sequences as input and generates two sequence as output:

$$Y_X, Y_P = \text{LunaAttn}(X, P, C) \tag{5}$$

where the computation of $Y_P$ and $Y_X$ is in (3) and (4). By stacking multiple layers of Luna attention, the output $Y_P$ from the previous layer, which captures contextual information of $C$, is employed as

the input $P$ of the next layer. For the first layer of Luna, we formulate $P$ as learnable positional embeddings[2] (Vaswani et al., 2017).

**Reducing the Number of Parameters.** Due to the two nested attention operations, there are two sets of parameters ($W_Q$, $W_K$, $W_V$) in a single Luna attention function. There are several techniques to reduce the number of parameters, such as parameter sharing (Xia et al., 2019). In this work, we follow Wang et al. (2020) to share $W_K$ and $W_Q$ in each layer, and conduct experiments to analyze performance decline against Luna with full sets of parameters (§4.2).

## 3.2 Luna Layers

The Luna attention is used as a drop-in-replacement for the regular attention. We incorporate the position-wise feed-forward network and layer normalization into Luna layers. Concretely, layer normalization is applied to both $Y_X$ and $Y_P$, while FFN only to $Y_X$:

$$
\begin{aligned}
Y_X, Y_P &= \text{LunaAttn}(X, P, C) \\
X_A, P_A &= \text{LayerNorm}(Y_X + X), \ \text{LayerNorm}(Y_P + P) \\
X', P' &= \text{LayerNorm}(\text{FFN}(X_A) + X_A), \ P_A
\end{aligned}
\tag{6}
$$

where $X'$ and $P'$ are the two outputs of the Luna layer. The graphical specification of one Luna layer is illustrated in Figure 2.

## 3.3 Luna Causal Attention

As discussed in Tay et al. (2020b), the ability to support causal autoregressive decoding, i.e. attending solely to the past and current tokens, is required when designing efficient self-attention mechanisms. However, due to the pack attention that packs the long sequence $X$ into a fixed (shorter) length, it is not straight-forward to support causal attention in Luna.

To design causal attention in Luna, we need to assume that the input $P$ contains no information of $X$, i.e. $P$ will not leak any future information of $X$ to the history. Before we describe the Luna causal attention mechanism, we first define a causal function $f : \mathbb{R}^{n \times d_1} \times \mathbb{R}^{n \times d_1} \times \mathbb{R}^{n \times d_2} \to \mathbb{R}^{n \times d_2}$:

$$
F \triangleq f(X, Y, Z), \ \text{where} \ F_t = \frac{1}{t} X_t \sum_{j=1}^{t} Y_j^T Z_j
\tag{7}
$$

where $F \in \mathbb{R}^{n \times d_2}$ and $F_t$ denotes the $t$-th row of $F$. From the definition of $f$ in (7), we see that $F_t$ can only access the information of the past and present row of $X$, $Y$ and $Z$.

To perform Luna causal attention, we first compute the attention matrix of the pack attention: $A_{pack} = \omega(PX^T/\sqrt{d})$. For simplicity, we omit the learnable parameters, e.g. $W_Q$, $W_K$, $W_V$ in (1). Note that for $A_{pack}$, we cannot use the softmax function for $\omega$, as the normalization term in softmax leaks future information of $X$ to the history. Inspired by the causal attention mechanism in Linear Transformer (Katharopoulos et al., 2020), we use two activation functions: 1) $\omega(\cdot) = \text{elu}(\cdot) + 1$ based on the exponential linear unit (Clevert et al., 2016); 2) $\omega(\cdot) = \text{softplus}(\cdot)$ based on the *softplus* function (Glorot et al., 2011). With the causal function $f$ in (7), we compute the attention matrix of the unpack attention: $A_{unpack} = \omega(f(X, X, A_{pack}^T))$. Unlike $A_{pack}$, we can use $\omega(\cdot) = \text{softmax}(\cdot)$ for $A_{unpack}$, because the normalization is along the $l$-dimension rather than the $n$-dimension of $X$. Finally, the output Y is computed by $Y = f(A_{unpack}, A_{pack}^T, X)$.

The complexity of the causal attention in Luna is still linear: $O(ln)$. One drawback of Luna causal attention, similar to the causal attention in Random Feature Attention (RFA) (Peng et al., 2021) and Linear Transformer (Katharopoulos et al., 2020), is its sequential computation for each timestep $t$.

**The sources of $P$.** In the formulation of causal attention, $P$ is expected to contain no information about $X$. Thus, we need to formulate $P$ based on the usage mode of the causal attention. For the *encoder-decoder* mode in sequence-to-sequence modeling (e.g. for machine translation), we can use packed output from the Luna encoder as $P$. For the *decoder-only* mode (e.g. for language modeling), $P$ might be formulated as a learnable parameter of each layer.

---

[2]We also experimented with sinusoidal positional embeddings, and obtained similar results.

Table 1: Experimental results on the long range arena (LRA) benchmark. For Luna, we explore three projected dimensions: 16, 128 and 256. 'Avg. (w/o rtl)' denotes the averaged accuracy over all tasks excluding Retrieval. The performance of previous works are from Tay et al. (2021).

| Models | ListOps | Text | Retrieval | Image | Pathfinder | Avg. | Avg. (w/o rtl) |
|---|---|---|---|---|---|---|---|
| Transformer | 36.37 | 64.27 | 57.46 | 42.44 | 71.40 | 54.39 | 53.62 |
| Transformer (re-impl) | 37.11 | 65.21 | 79.14 | 42.94 | 71.83 | 59.24 | 54.27 |
| Local Attention | 15.82 | 52.98 | 53.39 | 41.46 | 66.63 | 46.06 | 44.22 |
| Sparse Trans. | 17.07 | 63.58 | 59.59 | 44.24 | 71.71 | 51.24 | 49.15 |
| Longformer | 35.63 | 62.85 | 56.89 | 42.22 | 69.71 | 53.46 | 52.60 |
| Linformer | 35.70 | 53.94 | 52.27 | 38.56 | 76.34 | 51.36 | 51.14 |
| Reformer | 37.27 | 56.10 | 53.40 | 38.07 | 68.50 | 50.67 | 49.99 |
| Sinkhorn Trans. | 33.67 | 61.20 | 53.83 | 41.23 | 67.45 | 51.39 | 50.89 |
| Synthesizer | 36.99 | 61.68 | 54.67 | 41.61 | 69.45 | 52.88 | 52.43 |
| BigBird | 36.05 | 64.02 | 59.29 | 40.83 | 74.87 | 55.01 | 53.94 |
| Linear Trans. | 16.13 | 65.90 | 53.09 | 42.34 | 75.30 | 50.55 | 49.92 |
| Performer | 18.01 | 65.40 | 53.82 | 42.77 | 77.05 | 51.41 | 50.81 |
| Luna-16 | 37.43 | 65.74 | 79.38 | 46.39 | 78.36 | 61.46 | 56.98 |
| Luna-128 | **38.01** | 65.74 | 79.55 | 47.47 | **78.89** | 61.93 | 57.53 |
| Luna-256 | 37.98 | **65.78** | **79.56** | **47.86** | 78.55 | **61.95** | **57.54** |

## 3.4 Discussion

**Relation to Linformer and Shared Workspace.** One previous work closely related to Luna is Linformer (Wang et al., 2019). Linformer linearly projects the context sequence $C \in \mathbb{R}^{m \times d}$ into a sequence with fixed length $l$: $C' = EC$, where $C' \in \mathbb{R}^{l \times d}$ is the projected context sequence and $E \in \mathbb{R}^{l \times m}$ is the learnable projection matrix of each layer. Then, the attention operation is applied on the query $X$ and the projected context $C'$. The pack attention in Luna is a generalization of the linear projection in Linformer. There are two main advantages to Luna over Linformer: i) with pack attention as the projection method, Luna is able to model sequences with various lengths. In contrast, Linformer requires the length of all input sequences to be the same $m$, due to the projection matrix $E$, whose shape depends on $m$. ii) Luna achieves better expressiveness than Linear, not only due to the general projection method but also by encoding adequate contextual information into the projection via $P$ (see §3.1). Experimental improvements over non-contextual projection demonstrate the effectiveness of Luna (see §4.2). In contemporaneous and individual work, Goyal et al. (2021) formulate contextual $p$ as a shared global workspace, which shares similar instantiation with Luna.

**Relation to Set Transformer.** The additional input $P$ in Luna can be regarded as a side memory module that can access the entire sequence to gather contextual information. From this view of point, Luna is also closely related to Set Transformer (Lee et al., 2019), an early model to integrate side memory module in Transformers. Similar to the projection matrix in Linformer, the *inducing points* in Set Transformer are learnable parameters. Thus, these inducing points might be formulated as the non-contextual version of $P$ in Luna. Moreover, Set Transformer is designed for *set-input* problems, which are problems wherein the input is a set of features and the model is thereby invariant to permutation or ordering of the input features (Tay et al., 2020b), while Luna attention is used as a drop-in replacement for regular softmax attention.

# 4 Experiments

## 4.1 Long-Context Sequence Modeling

We evaluate the effectiveness and efficiency of Luna on the Long Range Arena (LRA) benchmark recently introduced by Tay et al. (2021), which is designed for the purpose of evaluating efficient Transformer models under the long-context scenario. They collect five tasks in this benchmark which are ListOps (Nangia and Bowman, 2018), byte-level text classification (Text; Maas et al., 2011), byte-level document retrieval (Retrieval; Radev et al., 2013), image classification on sequences of pixels (Image; Krizhevsky et al., 2009) and Pathfinder (Linsley et al., 2018). These tasks consist of input sequences ranging from 1K to 8K tokens and span across a variety of data types and modalities.

Table 2: Training speed and peak memory consumption comparison of different models on byte-level text classification with various input lengths (1K, 2K, 3K and 4K). The best model is in boldface.

| Model | Steps per second ↑ | | | | Peak Memory Usage (GB) ↓ | | | |
|---|---|---|---|---|---|---|---|---|
| | 1K | 2K | 3K | 4K | 1K | 2K | 3K | 4K |
| Transformer | 1.0 | 1.0 | 1.0 | 1.0 | 1.00 | 1.00 | 1.00 | 1.00 |
| Local Attention | 1.1 | 1.7 | 3.2 | 5.3 | 0.49 | 0.29 | 0.19 | 0.14 |
| Linformer | **1.2** | **1.9** | 3.7 | 5.5 | **0.44** | **0.21** | 0.18 | **0.10** |
| Reformer | 0.5 | 0.4 | 0.7 | 0.8 | 0.56 | 0.37 | 0.28 | 0.24 |
| Sinkhorn Trans | 1.1 | 1.6 | 2.9 | 3.8 | 0.55 | 0.31 | 0.21 | 0.16 |
| Synthesizer | 1.1 | 1.2 | 2.9 | 1.4 | 0.76 | 0.75 | 0.74 | 0.74 |
| BigBird | 0.9 | 0.8 | 1.2 | 1.1 | 0.91 | 0.56 | 0.40 | 0.30 |
| Linear Trans. | 1.1 | **1.9** | 3.7 | 5.6 | **0.44** | 0.22 | **0.15** | 0.11 |
| Performer | **1.2** | **1.9** | **3.8** | **5.7** | **0.44** | 0.22 | **0.15** | 0.11 |
| Luna-16 | **1.2** | 1.8 | 3.7 | 5.5 | **0.44** | 0.23 | 0.17 | **0.10** |
| Luna-128 | 1.1 | 1.7 | 3.4 | 5.1 | 0.49 | 0.28 | 0.21 | 0.14 |
| Luna-256 | 1.1 | 1.7 | 3.3 | 4.9 | 0.60 | 0.33 | 0.23 | 0.16 |

To ensure fair comparisons, for all tasks except for the task Retrieval, we closely follow the model configurations in Tay et al. (2021) such as data preprocessing, data split, model architecture, etc. For the task of Retrieval, we find that models are not fully converged when being trained for 5K steps as stated in Tay et al. (2021). Therefore, we train models for 20K steps for this task and obtain much better results. For a direct comparison, besides the average performance of models across all tasks, we also report the average accuracy on tasks excluding Retrieval. We run each experiment for five times with different random seeds and report the average accuracy. The hyper-parameters for each task are shown in Appendix A.1.

**Results.** The results of various models on the LRA benchmark are presented in Table 1. For our proposed method, we report results from models of three different projected dimensions (16, 128 and 256). First, we note that Luna achieves good results on all tasks consistently compared to the Transformer model and significantly outperforms all the other baseline methods in terms of the average accuracy. By taking a closer look at the accuracy for each individual task, Luna wins over baseline models on three out of five tasks and performs comparably with the best performed model on the other two tasks, i.e. ListOps and byte-level text classification. Notably, Luna improves over the Transformer model on image classification and pathfinder by a large margin. Second, we observe that although Luna achieves the best average performance with a projection dimension of 256, it also performs considerably well with smaller projection dimensions (16 and 128). This demonstrates the effectiveness of Luna even with small projected dimensions.

**Memory and Speed Efficiency.** Luna employs two nested linear attention functions to reduce the time and memory complexity compared to the vanilla softmax attention. Here, we examine the speed and memory footprint of various models with varying input lengths (1K, 2K, 3K and 4K). Following Tay et al. (2021), all models are evaluated on the byte-level classification task with the same batch size. The result is shown in Table 2.

Considering the memory efficiency, Luna with a projected dimension of 16 is highly memory-efficient, which is only 10% of the vanilla Transformer at 4K input sequence length. With larger projected dimensions, i.e. 128 and 256, Luna requires more memory but is still competitive compared to other efficient Transformer models. In terms of time efficiency, Luna-16 speeds up over the standard Transformer by 1.2-5.5 times, varying by the sequence length. Compared to other efficient Transformers, Luna-16 performs comparably with the fastest models, i.e. Performer and Linformer. Overall, our models achieve competitive advantage both in time- and memory-efficiency over other models, while attaining the best performance on the LRA benchmark (see Figure 1).

In addition, we plot the trade-off among memory, time and averaged LRA score without task Retrieval in Figure 1. Models such as Linformer and Performer have faster speed and small memory requirement with the sacrifice of performance. However, besides competitive time- and memory-efficiency, Luna models retain superior performance even with a small projected dimension ($l$=16).

**Contextual information in $P$ of Luna.** Recently, a popular method to model the classification task using Transformer-based models is to prepend a special symbol, [CLS], to every input example. The last hidden state of this symbol is regarded as the aggregate sequence representation. In Luna, we introduce an extra model input $P$ which not only allows us to efficiently compute the attention mechanism but learn contextual information as well. Theoretically, the $P$ from the last layer is capable of learning the representation of the input sequence. To validate this, we extract $P$ at the last layer and employ the mean pooling strategy over positions to obtain the final feature for classification. We test its performance on three long-text modeling tasks in LRA (Tay et al., 2021), i.e., ListOps, Text and Retrieval and report results in Table 3. We find that $P$-based methods obtain better scores across all tasks against the [CLS]-based one, validating the powerful ability of $P$ to encode contextual information of the input sequence.

Table 3: Performance comparison of two sentence representation methods on LRA benchmark.

| Models | ListOps | Text | Retrieval | Avg. |
|---|---|---|---|---|
| Luna-16, [CLS] | 37.43 | 65.74 | 79.38 | 60.85 |
| Luna-16, $P$ | 38.06 | 65.81 | 80.22 | 61.36 |
| Luna-128, [CLS] | 38.01 | 65.74 | 79.55 | 61.10 |
| Luna-128, $P$ | 38.27 | 65.89 | 80.27 | 61.48 |
| Luna-256, [CLS] | 37.98 | 65.78 | 79.56 | 61.11 |
| Luna-256, $P$ | 38.36 | 66.07 | 80.25 | 61.56 |

## 4.2 Machine translation

To evaluate Luna on sequence-to-sequence modeling, we conduct experiments on a standard machine translation benchmark, i.e. WMT'14 English-German (EN→DE) dataset (4.5M sentence pairs). The data split and preprocessing steps follow those of Vaswani et al. (2017), using the scripts from FairSeq (Ott et al., 2019). We share the source and target vocabularies within the language pair, with 37K byte pair encoding (BPE) types (Sennrich et al., 2016). The Luna models closely follow the architecture of Transformer-base: 6 encoder and decoder layers with 8 attention heads and $d_{\text{model}}/d_{\text{hidden}} = 512/2048$. We train the Transformer-base model with two optimization methods: Adam (Kingma and Ba, 2015) and Apollo (Ma, 2020), and find Apollo achieves better performance. Therefore, we use Apollo as the optimizer for all Luna models. For each experiment, we conduct distributed training across eight NVIDIA Tesla V100 GPUs with maximum batch size of 8192 tokens per GPU. Further details are provided in Appendix A.2.

Table 4: Test BLEU on WMT'14 EN→DE.

| Model | BLEU | # Param. |
|---|---|---|
| Transformer-base (Adam) | 27.8 | 64.9M |
| Transformer-base (Apollo) | 28.3 | 64.9M |
| RFA ($k = 256$) | 27.2 | 66.2M |
| Luna-16, elu, tied kv | 27.1 | 69.6M |
| Luna-32, elu, tied kv | 27.3 | 69.7M |
| Luna-16, softplus, tied kv | 27.3 | 69.6M |
| Luna-32, softplus, tied kv | 27.5 | 69.7M |
| Luna-16, elu | 27.4 | 77.5M |
| Luna-32, elu | 27.6 | 77.6M |
| Luna-16, softplus | 27.6 | 77.5M |
| Luna-32, softplus | 27.8 | 77.6M |

**Results.** Table 4 presents the results of Luna on the test set BLEU scores of WMT'14 EN→DE, along with Transformer-base and Random Feature Attention (RFA) as baselines. Different from Peng et al. (2021) where the random feature attention is applied only to decoders, the RFA model in Table 4 applies random feature attention in both the encoder and decoder for a fair comparison. $k = 256$ is the number of feature maps in RFA. For Luna, we report performance of models with different projected lengths: $l = 16$ and $l = 32$, different activation functions in (7): $\text{elu}(\cdot) + 1$ and $\text{softplus}(\cdot)$, and w./w.o parameter sharing.

From Table 4, the first observation is that $\text{softplus}(\cdot)$ consistently outperforms $\text{elu}(\cdot) + 1$. Thus, we use $\text{softplus}(\cdot)$ as the default activation function in the implementation. Another interesting observation is that Luna with a small projected length ($l = 16$) obtains similar performance to RFA with $k = 256$ feature maps. Luna with $l = 32$ achieves competitive performance, but still falls behind the Transformer-base model. Further improving the machine translation performance of Luna is left to future work. We also report the number of parameters of different models. At last, we evaluate Luna w./w.o parameter sharing. Although there are two sets of parameters in a single Luna attention function ($W_Q$, $W_K$, $W_V$), as mentioned in §3.1, we tie $W_k$ with $W_v$ to reduce the number of parameters, and the performance decline is marginal. As a result, Luna with shared parameters has 7% and 5% more parameters compared to the vanilla Transformer and RFA models.

**Effect of Encoding Contextual Information into $P$.** As discussed in §3.4, one advantage of Luna against Linformer is to incorporate contextual $P$ by formulating it as an extra input. To investigate the importance of this design, we conduct experiments on WMT'14 to compare Luna with the baseline model where $P$ is formulated as a non-contextual learnable parameter of each layer.

Table 5: Dev and Test BLEU

| Model | Dev. | Test |
|---|---|---|
| Non-Contextual | 24.4 | 25.2 |
| Contextual | **25.9** | **27.3** |

For both the contextual and non-contextual models, we train Luna with $l = 16$, parameter sharing and softplus. Table 5 lists the BLEU scores on the development and test sets. Luna with contextual $P$ significantly outperforms the baseline with non-contextual $P$, demonstrating the effectiveness of this design in Luna.

### 4.3 Masked Language Modeling for Large-Scale Pretraining

One popular application of Transformer is to pretrain a large-scale language model on a large amount of data which can then be fine-tuned on a wide range of downstream tasks, such as BERT (Devlin et al., 2019), RoBERTa (Liu et al., 2019), etc. Therefore, we pretrain a Luna-based language model with RoBERTa-base model configuration on two versions of data as our pretraining set: 1) BERT version with BookCorpus (Zhu et al., 2015) and English Wikipedia (totally 16GB), 2) RoBERTa version with BookCorpus, English Wikipedia, CC-News (Nagel, 2016), OpenWebText (Gokaslan and Cohen, 2019) and Stories (Trinh and Le, 2018) (totally 160GB). For Luna models, we set $l = 128$. On the larger training corpus (160GB), we train models w./w.o parameter sharing, respectively. We compare our models with RoBERTa-base, BERT-base and Linformer which are trained on the same training data. Experimental details are provided in Appendix A.3.

**Finetuning Luna** After obtaining the pretrained Luna-based language model, we finetune it on various natural language processing tasks, including sentiment classification (SST-2; Socher et al., 2013), natural language inference (QNLI; Rajpurkar et al., 2016), textual similarity (QQP; Chen et al., 2018, question answering (RACE (Lai et al., 2017) and CommonsenseQA (CSQA; Talmor et al., 2019). For GLUE tasks, following Liu et al. (2019), we consider a limited hyperparameter sweep for each task, with batch sizes $\in \{16, 32\}$ and learning rate $\in \{5e^{-6}, 1e^{-5}, 2e^{-5}\}$, with a linear warmup for the first 6% of steps followed by a linear decay to 0. Finetuning is performed for 20 epochs with early stopping based on each task's evaluation metric on the dev set[3]. For QA tasks, we concatenate each candidate answer with the corresponding question and passage. We then encode every candidate and pass the [CLS] output at the last layer through a fully-connected layer, which is used to predict the correct answer. We truncate question-answer pairs that are longer than 128 tokens and, if needed, the passage so that the total length is at most 512 tokens. Following Liu et al. (2019), we try a small range of possible values for hyperparameters, i.e., batch size $\in \{16, 32\}$, learning rate $\in \{1e^{-5}, 2e^{-5}, 3e^{-5}\}$ and dropout $\in \{0.0, 0.1, 0.2\}$. For other configurations such as warm-up steps, optimizer, we follow thoses in Liu et al. (2019).

The result is reported in Table 6. We observe that on the smaller dataset (16GB) our Luna model has similar or slightly better downstream results compared to other pretrained language models. On QNLI and SST-2, Luna models obtain the best performance among all models, reaffirming the effectiveness of Luna in pre-training. This demonstrates the strong ability of Luna for language representations. On the larger dataset (160GB), however, the performance of Luna is slightly worse than RoBERTa with vanilla Transformer architecture. One possible reason is that the capacity of Luna is not as sufficient as vanilla Transformer, due to the efficient attention mechanism. This is supported by the evidence that Luna with full sets of parameters achieves better performance than that with parameter-sharing, because Luna with full sets of parameters has better capacity.

## 5   Related Work

There has been signficiant prior work on improving the efficiency of Transformers, besides the three closely related works discussed in §3.4. The common techniques include, but are not limited to, weight sharing (Dehghani et al., 2018), quantization (Shen et al., 2020; Fan et al., 2020), sparse attention (Parmar et al., 2018; Kitaev et al., 2020), side memory module (Lee et al., 2019; Gupta and Berant, 2020; Goyal et al., 2021), and low-rank or compressed context (Wang et al., 2019; Ainslie

---

[3]We observed that Luna finetuning requires more epochs than vanilla Transformer (20 vs. 10). We also finetuned RoBERTa with 20 epochs but did not obtain better results.

Table 6: Performance of various models on development set of benchmark natural language understanding tasks. Bold face indicates best performance.

| Model | data | GLUE | | | QA | |
| --- | --- | --- | --- | --- | --- | --- |
| | | SST-2 | QNLI | QQP | RACE | CSQA |
| BERT-base | 16GB | 92.7 | 88.4 | 89.6 | 64.2 | **53.3** |
| RoBERTa-base | 16GB | **93.1** | 90.9 | **90.9** | **65.6** | - |
| Linformer-128 | 16GB | 92.4 | 90.4 | 90.2 | - | - |
| Luna-128, tied kv | 16GB | **93.1** | **91.2** | 90.8 | 65.2 | 53.1 |
| RoBERTa-base | 160GB | **94.8** | **92.8** | **91.9** | **73.50** | **63.61** |
| Luna-128, tied kv | 160GB | 94.3 | 91.5 | 91.2 | 71.50 | 61.48 |
| Luna-128 | 160GB | 94.6 | 92.2 | 91.3 | 72.25 | 62.08 |

et al., 2020). In this section, we briefly review some recently proposed methods. For a detailed overview we refer the readers to Tay et al. (2020b).

**Sparse Attention** The general idea of these methods is that, instead of attending to the whole sequence, each token only access to a fixed, predefined range such as local neighborhoods and strided or "dilated" windows. Popular methods include local attention (Parmar et al., 2018), blockwise attention (Qiu et al., 2019), strided attention patterns (Child et al., 2019; Beltagy et al., 2020), and compressed attention (Liu et al., 2018). To make this range more flexible, Reformer (Kitaev et al., 2020) employs a hash-based similarity measure to efficiently cluster tokens into chunks and Routing Transformer(Roy et al., 2021) employ online k-means clustering on the tokens. The Sinkhorn sorting Network (Tay et al., 2020a) exposes the sparsity in attention weights by learning to sort blocks of the input sequence.

**Kernel Methods.** A recently popular method to improve the efficiency of Transformers is to avoid explicitly computing the $m \times n$ attention matrix $A$ in (1) by re-writing it with kernels. Typical models leveraging kernelization are Linear Transformer (Katharopoulos et al., 2020), Performer (Choromanski et al., 2020) and Random Feature Attention (Peng et al., 2021). Since kernels are a form of approximation of the attention matrix, they can be also viewed as a form of low-rank method (Choromanski et al., 2020) that compresses the context to a shorter length, such as Linformer (Wang et al., 2019) and the proposed Luna model.

**Recurrence.** The simplest technique to reduce the complexity of Transformer is to chunk input sequences into fixed blocks, with the obvious disadvantage of losing contextual information from past chunks. As discussed in Tay et al. (2020b), these models can be regarded as *fixed pattern* models. Transformer-XL (Dai et al., 2019) proposed a natural extension to the blockwise method to connect these blocks via a recurrence mechanism. Compressive Transformer (Rae et al., 2020) further extends Transformer-XL by maintaining a fine-grained memory of past chunk activations, which are discarded in Transformer-XL. Technically, Luna can be adapted to a recurrence method, by simply using $P$ as an inherent memory module to maintain the recurrence across segments.

# 6 Conclusion

We have introduced Luna, a simple, efficient and effective linear attention mechanism used as a drop-in substitute for regular softmax attention. By introducing an extra input with the fixed length, Luna is capable of capturing adequate contextual information while performing attention operations linearly. On three sequence modeling tasks, i.e., long-context sequence modeling, neural machine translation, and large-scale pretraining and finetuning, Luna achieves comparable or even better performance than a variety of strong baselines, while acquiring prominent gains of efficiency in both speed and memory. In future work, we are interested in combining Luna with recurrence methods where $P$ can be used as a running memory across segments of inputs. Another interesting direction would be to apply Luna to other tasks with long input sequences, such as document-level summarization and translation.

## Acknowledgments and Disclosure of Funding

This material is based on research sponsored by Air Force Research Laboratory (AFRL) under agreement number FA8750-19-1-1000. The U.S. Government is authorized to reproduce and distribute reprints for Government purposes notwithstanding any copyright notation therein. Xiang Kong was supported by U.S. DARPA AIDA Program No. FA8750-18-2-0014. The views and conclusions contained herein are those of the authors and should not be interpreted as necessarily representing the official policies or endorsements, either expressed or implied, of Air Force Laboratory, DARPA or the U.S. Government.

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
