# Appendix: Luna: Linear Unified Nested Attention

## A  Experimental Details

### A.1  Long-Context Sequence Modelling

For all tasks except Retrieval, we closely follow the model configurations in Tay et al. (2021) such as data preprocessing, data split, model architecture, batch size etc. To guarantee convergence, we train models for the Retrieval task with 20k steps instead of the 5k steps prescribed inTay et al. (2021). The hyperparameters of models in these tasks are listed in Table 7. We mainly tune three hyperparameters: learning rate, dropout and attention dropout. For the other main hyperparametrs such as batch size, number of layers and number of warmup steps, we follow the guidance of Tay et al. (2021).

Table 7: Hyperparameters of models in LRA tasks. LR and Attn-Dropout denote the learning, batch size and attention dropout.

| Tasks | LR | Dropout | Attn-Dropout |
|---|---|---|---|
| ListOps | 1e-4 | 0.1 | 0.1 |
| Text | 5e-5 | 0.3 | 0.3 |
| Retrieval | 5e-5 | 0.1 | 0.1 |
| Image | 5e-3 | 0.1 | 0.3 |
| Pathfinder | 1e-3 | 0.2 | 0.1 |

### A.2  Neural Machine Translation

Our experiments on WMT 2014 English-German are based on the Transformer-base model (Vaswani et al., 2017), with implementation from the FairSeq package (Ott et al., 2019). This dataset contains 4.5M parallel sentence pairs for training. We following the standard setting (Vaswani et al., 2017), using Newstest2013 as the validation set and Newstest2014 as the test set. The dataset is pre-processed following (Ma, 2020), using the scripts from FairSeq package[4]. Specifically, we use word embedding with 512 dimension and 6-layer encoder/decoder with 8 multi-head attention and 2048 feed-forward dimensions. We apply 0.1 label smoothing (Szegedy et al., 2016), and perform totally $500,000$ updates to train each model. For Adam, we use start learning rate $0.0005$, set $\beta = (0.9, 0.98)$, and apply the decoupled weight decay technique (AdamW) (Loshchilov and Hutter, 2019). For all the models trained with APOLLO, we set the learning rate is $0.1$, $\beta = 0.9$ and $\epsilon = 1e^{-4}$. For learning rate scheduling, we applied linear warm up the learning rate for both Adam, and APOLLO — $4000$ updates for Adam and $1000$ updates and APOLLO. After learning rate warming up, we applied the inverse square root decay (Vaswani et al., 2017) to Adam. For APOLLO, following Ma (2020), we decayed the learning rate at the $300,000$ and $450,000$ updates by decay rate $0.1$. Gradient clips with $1.0$ are applied to all the optimization methods, and the dropout ratio are set to $0.1$. Weight decay rates are $1e^{-4}$ for Adam methods and $1e^{-8}$ for APOLLO. The decoding beam size is set to $5$, and the checkpoints of the last 10 epochs are averaged before evaluation. For each experiment, we conducted distributed training across eight NVIDIA Tesla V100 GPUs with maximum batch size as 8192 tokens per GPU (totally $8192 \times 8$ tokens per batch).

### A.3  Masked Language Modeling for Large-Scale Pretraining and Finetuing

We pre-trained all the models on 64 Tesla V100 GPUs with the standard masked-language-modeling (MLM) objective and two pre-training corpus: (i)BERT version with BookCorpus (Zhu et al., 2015) and English Wikipedia (totally 16GB); (ii) RoBERTa version with BookCorpus, English Wikipedia, CC-News (Nagel, 2016), OpenWebText (Gokaslan and Cohen, 2019) and Stories (Trinh and Le, 2018) (totally 160GB). We use the standard Adam optimizer with a linear decay learning rate scheduler. Table 8 describes the hyperparameters for pre-training of Luna-128 model. For finetuning stage, we closely follow the training configuration used in released Roberta finetuning script for different tasks and main hyperparameters are listed in Table 9.

---

[4]https://github.com/pytorch/fairseq

Table 8: Hyperparameters for pre-training LUNA-128 on two public corpus.

| Hyperparameter | LUNA (16GB) | LUNA (160GB) |
|---|---|---|
| Number of Layers | 12 | 12 |
| Hidden size | 768 | 768 |
| FFN inner hidden size | 3072 | 3072 |
| Attention heads | 12 | 12 |
| Attention head size | 64 | 64 |
| Dropout | 0.1 | 0.1 |
| Attention Dropout | 0.1 | 0.1 |
| Warmup Steps | 15k | 24k |
| Peak Learning Rate | 6e-4 | 6e-4 |
| Batch Size | 2k | 8k |
| Weight Decay | 0.01 | 0.01 |
| Max Steps | 250K | 500k |
| Learning Rate Decay | Linear | Linear |
| Adam $\epsilon$ | 1e-6 | 1e-6 |
| Adam $\beta_1$ | 0.9 | 0.9 |
| Adam $\beta_2$ | 0.98 | 0.98 |
| Gradient Clipping | 1.0 | 1.0 |
| Project Length | 128 | 128 |

Table 9: Hyperparameters for finetuning Luna on GLUE, RACE and CSQA.

| Hyperparameter | GLUE | RACE | CSQA |
|---|---|---|---|
| Learning Rate | 1e-5 | 1e-5 | 1e-5 |
| Batch Size | 32 | 64 | 64 |
| Weight Decay | 0.1 | 0.01 | 0.01 |
| Max Epochs | 20 | 20 | 20 |
| Learning Rate Decay | Linear | Fixed | Polynomial Decay |
| Warmup Steps | 6% | 150 | 150 |
| Dropout | 0.1 | 0.1 | 0.2 |
| Attention Dropout | 0.1 | 0.1 | 0.0 |
| Activation Dropout | 0.1 | 0.0 | 0.1 |