# OpenReview forum: "Luna: Linear Unified Nested Attention"
_NeurIPS.cc/2021/Conference — NeurIPS 2021 Poster_

### Official Review · Reviewer_Lezq · 2021-07-12

**Rating:** 6
**Confidence:** 4

**Summary:**

This paper proposes Luna, an approximation to the Transformer's softmax attention with linear (in sequence length) time and space complexity. The approximation works using two attention functions. The first packs the sequence into one of fixed length; the second unpacks this fixed length sequence into one whose length matches the original sequence. Experiments show that Luna is computationally efficient, and typically performs either competitively or better than baseline methods.

**Limitations And Societal Impact:**

Besides the fact that Luna causal attention requires sequential computation for each time step, limitations of the method are not presented.

**Main Review:**

This is a clearly written piece of work. The approach is novel and it is made clear how it differs from prior work. The method is a simple and elegant way to reduce both the time and space complexity of the Transformer's attention. The experiments show significant computational improvements (particularly for long sequences) over the naive attention mechanism, and are competitive against other efficient Transformer baselines. The empirical results are also strong, with Luna performing either competitively with or better than baselines on a wide range of tasks.

However, the experiments section in particular leaves some room for improvement:
- On several tasks, Luna outperforms the naive attention mechanism; this is surprising given that Luna is, in theory, an approximation to naive attention. It would therefore add value to the paper if some intuition behind these surprising results were provided.
- The models in Table 2 are evaluated on the byte-level classification task, so presumably they do not use the causal attention architecture. However in lines 152-154, a limitation of the Luna causal attention is presented, namely that it requires sequential computation for each time step. It would therefore be useful to readers if the training speed were shown for the causal archiecture too.
- In Section 4.2, certain hyperparameters for the pre-training procedure are not provided, e.g. batch size, learning rate.
- The final part of Section 4.1 shows that using the final $Y_P$ and mean-pooling works better than using CLS, so why not use this strategy for the tasks in Section 4.2?

**Time Spent Reviewing:**

5

---

> ### Author Response · Authors · 2021-08-10
> **Response to Review Lezq**
>
> Thanks for your comments and positive feedback!
> We respond below to your questions and comments. We kindly request that you briefly read these responses and let us know if they do not fully satisfy your concerns.
>
> > On several tasks, Luna outperforms the naive attention mechanism; this is surprising given that Luna is, in theory, an approximation to naive attention.
>
> The tasks on which Luna outperforms Transformer are mainly classification problems. We hypothesize that in the Luna model, the packing process could summarize key information in the input contexts, which may benefit the final classification accuracy. In tasks such as machine translation that needs to store more contextual information, the standard Transformer performs slightly better than Luna.
>
> > The limitations of speed of autoregressive Luna attention
>
> We appreciate your suggestion of providing speed comparison of autoregressive Luna attention. We provide some preliminary results here and will add detailed results in the revised version.
> First, we have not implemented CUDA kernel for autoregressive Luna attention, and the training time of Luna on machine translation is about 1.3 times slower than Transformer, due to the sequential computation in autoregressive Luna attention. However, we believe that this slowdown is mainly due to the overhead of PyTorch/CUDA communication, and we can achieve significant acceleration by implementing CUDA kernel for it.
>
> Fortunately, we can fairly compare decoding time of Luna. We conducted preliminary experiments on machine translation decoding on WMT English-German dataset with different lengths of source sentences. Compared with Transformer, Luna decoding is faster when the source sentences are longer than 64 tokens. For sentences with 32 to 64 tokens, the decoding speed is similar, and when shorter than 32 Transformer is faster.
>
> >In Section 4.2, certain hyperparameters for the pre-training procedure are not provided, e.g. batch size, learning rate.
>
> We appreciate your suggestion to provide these hyper-parameters clearly. Basically, we used the same hyper-parameters with the same experiments in Linformer. Concretely, we set batch-size$=2k$ and learning-rate$=6e-4$. We will add these hyper-parameters in the appendix of our revised version.
>
> > The final part of Section 4.1 shows that using the final $Y_p$ and mean-pooling works better than using CLS, so why not use this strategy for the tasks in Section 4.2?
>
> We did not use $Y_p$ on the tasks in section 4.2 for two reasons. First, due to limits of time and computational resources, we did not find an opportunity to comprehensively explore this on the tasks in section 4.2. Second, based on some preliminary results, using $Y_p$ did not obtain significant improvements on these tasks.

---

### Official Review · Reviewer_FUG1 · 2021-07-16

**Rating:** 7
**Confidence:** 5

**Summary:**

The authors propose a novel operation to replace full self attention by compressing the information of the input features to a smaller set of features using attention and then attending to this smaller set to generate the outputs. The authors provide experiments on synthetic benchmarks as well as real world NLP tasks showcasing the performance of the proposed attention operation and comparing it with various state-of-the-art efficient attention methods.

**Limitations And Societal Impact:**

I would like a little more discussion on the limitations of autoregressive Luna attention. The authors, judging from their submitted code, do not seem to use the CUDA kernels of (Katharopoulos et al, 2020). Although that takes away from maintaining a custom CUDA kernel, what is the speed comparison for autoregressive tasks such as decoding for machine translation?

No societal impact discussion is needed.

**Main Review:**

Strengths
--------------

- The idea of compressing the sequence using attention is simple and works very well
- The performance on LRA is impressive both in term of accuracy and speed
- The authors showcase competitive performance on commonly used datasets on machine translation and MLM

Weaknesses
-----------------

- Although LRA is an interesting benchmark the performance comparison in terms of accuracy is relatively weak. All models seem to have high variance in performance and if standard deviation was reported the numbers would carry significantly less importance. That said, Luna performs significantly better than other methods (however it also performs significantly better than full attention (3 percentage points)). This is also observed in MLM and machine translation for which the performance is much closer to other efficient attention models.
- Performance in terms of wall-clock time per epoch (or for the forward pass) is not provided for the real world datasets. For instance is Linformer faster in the MLM pretraining? Is RFA faster for machine translation? Also there is no performance comparison for the autoregressive Luna attention.

I believe that the above are not significant issues, however they make it harder to place Luna attention in the performance/speed spectrum for real world datasets or autoregressive tasks. For instance, how much faster is Luna for machine translation given that it achieves 1 BLEU point lower performance than a traditional transformer.

### Minor points

- Although distinctly different, I believe that the idea is sufficiently related to Perceiver [1] such that it warrants a discussion let alone a mention.

[1]: Perceiver: General Perception with Iterative Attention

**Time Spent Reviewing:**

8

---

> ### Author Response · Authors · 2021-08-10
> **Response to Review FUG1**
>
> Thanks for your comments and positive feedback!
> We respond below to your questions and comments. We kindly request that you briefly read these responses and let us know if they do not fully satisfy your concerns.
>
> > The limitations of speed of autoregressive Luna attention
>
> We appreciate your suggestion of providing speed comparison of autoregressive Luna attention. We provide some preliminary results here and will add detailed results in the revised version.
> First, we have not implemented CUDA kernel for autoregressive Luna attention, and the training time of Luna on machine translation is about 1.3 times slower than Transformer, due to the sequential computation in autoregressive Luna attention. However, we believe that this slowdown is mainly due to the overhead of PyTorch/CUDA communication, and we can achieve significant acceleration by implementing CUDA kernel for it.
>
> Fortunately, we can fairly compare decoding time of Luna without CUDA kernel. We conducted preliminary experiments on machine translation decoding on WMT English-German dataset with different lengths of source sentences. Compared with Transformer, Luna decoding is faster when the source sentences are longer than 64 tokens. For sentences with 32 to 64 tokens, the decoding speed is similar, and when shorter than 32 Transformer is faster.
>
> > Although distinctly different, I believe that the idea is sufficiently related to Perceiver [1] such that it warrants a discussion let alone a mention.
>
> Thanks for pointing out the related work we missed. We will elaborate on them in the related work section of the revised version.

---

### Official Review · Reviewer_1DnM · 2021-07-16

**Rating:** 7
**Confidence:** 5

**Summary:**

The paper proposes a linear time and memory attention mechanism that computes attention into two steps: In first step, an extra input ($P$) with fixed sequence length is introduced that summarizes the context to fixed length output sequence via packing attention. In the next step, the queries attend (unpack attention) to the fixed length summarized sequence resulting in a overall complexity that is linear in input sequences length. Experiments are conducted on LRA benchmark and other NLP tasks to show the effectiveness of the proposed method against other fast attention mechanisms.

**Main Review:**

- **Originality**:

  - The paper presents a linear attention mechanism that first packs the context into a fixed length representation and then queries attend to the packed sequence to get the outputs. While for the self-attention, Set Transformers have been proposed using the same mechanism, the novelty of the paper lies in generalizing this mechanism to non-trivial causal auto-regressive modeling.  Moreover, as opposed to using learnable parameter to model the $P$ sequence for each layer, the paper also proposes to formulate $P$ as  learnable positional embedding for only first layer and the output of packed attention is fed as the input sequence $P$ to the next layer. This is later shown to have impact on the performance.

- **Quality**:

  - The paper is well organized, appropriately discusses the relation to other works, and discusses well different aspects of Luna attention such as the sources of $P$. I also appreciate the experiment validating the contextual information in $P$ as well as comparing the effect of encoding $P$ which is an important design parameter proposed in the work.
  - The experiments on LRA across other domains showcase the effectiveness of Luna against full attention and other fast attention mechanisms.
  - Experiments on the pre-training tasks and machine translation demonstrate the effectiveness on difficult large scale settings.

- **Clarity**:
  - The paper is well written and easy to follow. The paper discusses differences with closely related works such as Set Transformers and Linformer. Ablations analyzing contributions from different aspects of the work are also well analyzed and described. All details including the code should be enough to reproduce this work.

- **Significance**:
  - This paper tackles important problem of improving the computational efficiency of attention. The idea is well presented and generalizes the packing and unpacking attention to causal scenarios which is an important distinction with Set Transformers.  Experimental results on LRA show that compared to other fast attentions, Luna offers favorable trade-offs for time, memory, and performance benchmarks.

 - **Suggestions for improvement**:
   - Calling method "approximation" of softmax attention seems a bit misleading.  Approximating softmax gives a connotation of being backward compatible or attention output approximating softmax attention patterns. In the absence of any empirical results or theoretical arguments towards this, I would suggest to rephrase this more appropriately.

   - I would encourage the authors to include the following two relevant work in related works discussion:
      - "SMYRF: Efficient Attention using Asymmetric Clustering"
      - "Fast Transformers with Clustered Attention"

   - Discussion on limitations:
      - The paper currently is missing any discussion on potential limitations imposed by restricting the length of $P$. For instance, can the $P$ pose issues if a task requires to copy tokens from context and the number of tokens to be copied is more tokens than the $P$ length? Or would more layers can overcome this limitation? It would interesting to do some analysis on the sequence duplication synthetic task as defined in Reformer.

    - Visualization of information captured by $P$.
      - It would be interesting to see while packing to small context, how is attention distributed over the context tokens. A few visualization on the question answering tasks could be very insightful.

**Time Spent Reviewing:**

8-9 hours

---

> ### Author Response · Authors · 2021-08-10
> **Response to Review 1DnM**
>
> Thanks for your comments and positive feedback!
> We respond below to your questions and comments. We kindly request that you briefly read these responses and let us know if they do not fully satisfy your concerns.
>
> > Calling method "approximation" of softmax attention seems a bit misleading.
>
> We appreciate your suggestion to rephrase this and apologize for the confusion.
>
> > I would encourage the authors to include the following two relevant work in related works discussion
>
> Thanks for pointing out the related work we missed. We will elaborate on them in the related work section of the revised version.
>
> > It would interesting to do some analysis on the sequence duplication synthetic task as defined in Reformer.
>
> We appreciate your suggestion of the sentence duplication task. We follow the experiment and model setting of the sentence duplication task mentioned in Reformer. Specifically, the model input sequence is of 1024 length and 1-layer Luna with model dimension 256 and 4 attention heads is employed. All models are trained with 150K steps. The accuracies of Luna with four different projection lengths of $p$, i.e. 32, 64, 128 and 256, are presented in the following table:
>
> | Length of $p$ |  Accuracy |
> | :---------------   | :-----------: |
> | 32                  |  55.2%     |
> | 64                  |  97.1%     |
> | 128                |  99.9%     |
> | 256                |  100.0%   |
>
> This demonstrates that with large-enough projection lengths, Luna is able to pack global context of input sequence without loss of information.
>
> > Visualization of information captured by $p$
>
> Thanks for this valuable suggestion. We apply the Luna model with the project length 64 on the word-level IMDB sentiment analysis task. After checking the top-10 attention weights over the context tokens across different positions of $p$, we find that the same word in the context will be mainly attended by $p$ at the same position. For example in this sample “I can't believe how many ***good*** reviews I read on here. … The only ***good*** part was the first chase scene,...”
> The word ***good*** which occurs twice in this sample is attended by the same position in $p$ with large attention weights.
> Furthermore some semantically-similar words such as (***begin*** and ***start***), will also be grouped by the same position.
> These observations indicate that  the packing process is compressing similar information from the original context into the smaller one, which will largely retain the original information.

---

### Author Response · Authors · 2021-08-30
**Look forward to post-rebuttal feedback**

Dear Reviewers,

Thanks the reviewers for giving us a lot of constructive and valuable feedback, again!
We have posted detailed responses to your questions and concerns, and look forward to your post-rebuttal feedback and discussion.

---

### Decision · Program_Chairs · 2021-09-27

**Decision:**

Accept (Poster)

**Comment:**

This paper proposes a linear attention based on a pack and unpack mechanism of an input sequence. Experiments on LRA, pretraining and finetuning, and machine translation demonstrate the benefit of the proposed approach. I think this is a useful addition to the linear attention literature. Al reviewers generally agree that this is a good paper and provided some suggestions, which the authors mentioned will be added to the paper. I recommend accepting this paper.